# The Utilisation of Mushroom Leftovers, Oats, and Lactose-Free Milk Powder for the Development of Geriatric Formulation

**DOI:** 10.3390/foods13111738

**Published:** 2024-06-01

**Authors:** Snigdha Paul, Ravinder Kaushik, Shuchi Upadhyay, Ansab Akhtar, Prince Chawla, Naveen Kumar, Saurabh Sharma, Pooja Rani

**Affiliations:** 1School of Health Sciences and Technology, University of Petroleum and Energy Studies UPES, Bidholi, Dehradun 248007, India; paulsnigdha.sp@gmail.com (S.P.); shuchi.upadhyay@ddn.upes.ac.in (S.U.); 2School of Medicine, Louisiana State University, New Orleans, LA 70112, USA; ansabakhtar@gmail.com; 3Department of Food Technology and Nutrition, Lovely Professional University, Phagwara 144001, India; princefoodtech@gmail.com; 4Chitkara University Research and Innovation Network (CURIN), Chitkara University, Rajpura 140401, India; nkft87@gmail.com; 5General Surgery, School of Medicine, Stanford University, 300 Pasteur Drive, Palo Alto, CA 94305, USA; ssharma6@stanford.edu; 6Department of Commerce, Government College for Women, Gharaunda 132001, India; poojasharmakaushik1@gmail.com

**Keywords:** ergosterol, geriatric formulation, lactose free, mushroom leftovers, oats

## Abstract

This study aims to focus on developing a food supplement for the geriatric population using disposal mushrooms, oats, and lactose-free milk powder. Lactose intolerance is most common in older adults, raising the demand for lactose-free foods. One of the major global challenges currently faced by humankind is food waste (FW). Most of the food that is produced for human consumption has not been utilized completely (1/3rd–1/2 unutilized), resulting in agricultural food waste. Mushrooms are highly valuable in terms of their nutritional value and medicinal properties; however, a significant percentage of mushroom leftovers are produced during mushroom production that do not meet retailers’ standards (deformation of caps/stalks) and are left unattended. Oats are rich in dietary fibre beta-glucan (55% water soluble; 45% water insoluble). Lactose-free milk powder, oats, and dried mushroom leftover powder were blended in different ratios. It was observed that increasing the amount of mushroom leftover powder increases the protein content while diluting calories. The product with 15% mushroom powder and 30% oat powder showed the highest sensory scores and the lowest microbial count. The GCMS and FTIR analyses confirmed the presence of ergosterol and other functional groups. The results of the XRD analysis showed that the product with 15% mushroom powder and 30% oat powder had a less crystalline structure than the product with 5% mushroom powder and 40% oat powder and the product with 10% mushroom powder and 35% oat powder, resulting in more solubility. The ICP-OES analysis showed significant concentrations of calcium, potassium, magnesium, sodium, and zinc. The coliform count was nil for the products, and the bacterial count was below the limited range (3 × 10^2^ cfu/g). The product with 15% mushroom powder and 30% oat powder showed the best results, so this developed product is recommended for older adults.

## 1. Introduction

The progression of age being a multidimensional process leads to the deficiency of many nutrients due to many physiological, social, and physical changes in the body such as a reduction in enzyme activity, a reduction in appetite, an increase in nutritional needs, a reduction in detoxification potential, etc. [1,2]. Regarding the reduction in enzyme activity, studies have shown that with age, lactase enzyme activity, which helps digest milk lactose, decreases, causing older people to suffer from lactose intolerance, resulting in many problems like bloating, gas, discomfort, etc. [2,3,4]. Not only do physiological problems increase with age, but also the need for nutrients. One of them is fibre, which is very important for older people. A low consumption or intake of fibre in older people leads to constipation and other colon disorders, dysbiosis of the microbiome, and related problems including metabolic syndrome and obesity-induced inflammatory reactions [5]. Bone density loss is prevalent among older adults, increasing the risk of osteoporosis. Vitamin D_3_ and calcium are essential for maintaining bone health. Vitamin D_3_ deficiency in older adults results in the restriction of calcium re-absorption in the kidneys and raises parathyroid hormone secretion and the resorption of the bone [6]. Both the inability of the intestine to absorb vitamin D_3_ and the kidney’s inability to turn 25(OH) (vitamin D_3_) into 1.25(OH) (vitamin D) contribute to vitamin D3 insufficiency. To reduce the frequency of osteoporotic fractures, vitamin D_3_ and calcium levels must be maintained, which can be accomplished through dietary sources or supplementation [2]. The regular consumption of vitamin D_3_ is also linked to a lower mortality rate and less severe COVID-19 in hospitalized fragile older patients [7]. The consumption of foods that have essential nutrients required for older adults is very important to overcome deficiencies. Therefore, there is a greater need for the development of fortified foods enriched with vitamins for older adults [1].

If lactose from bovine milk is removed, it can be considered a source of protein and calcium supplements for the older population, as plant-based milk substitutes have some detrimental effects on health, such as issues due to a lack of protein, poor mineral and vitamin absorption, and dental health issues [8].

Oats have gained the label “super grain” due to the health advantages they offer. Oats and oat products include a variety of bioactive substances, including soluble fibre, polyphenols, and avenanthramides (antioxidants), which may have health advantages [9]. Khanna and Mohan [9] claimed that oats have an ability to decrease cholesterol, and the use of oats and oat-containing products that provide the required level of beta-glucan has been approved by the FDA. As it provides all of the mentioned advantages, especially richness in fibre, it can be used as a source of fibre in food products, which will be beneficial for older adults in many ways like easing the problem of constipation and other colon-related problems [5].

Mushrooms are a highly potential source of proteins, vitamins, dietary fibres, and minerals. According to Kumar et. al. [10], mushrooms are regarded as the safest and most comprehensive food; they are relevant for all age groups, versatile, and nutrient-rich. They can also be used in place of meat, fruit, fish, etc., especially for vegetarians. However, mushrooms also contain a sugar named trehalose, which is digested with the help of the enzyme trehalase in the intestine and can affect the digestion of trehalose [11], but trehalose intolerance is less prevalent than lactose intolerance, with only a few occurrences in the literature (<1%) [12]. Moreover, in healthy people, trehalose can stabilise blood sugar levels by causing a slower blood glucose release and a milder insulin response than other monosaccharides and disaccharides [13].

Despite being so valuable, mushrooms are not being utilized to their full potential. A lot of mushroom leftovers produced during mushroom production do not meet the retailer’s standards (deformation of caps/stalks) and are unattended. According to Murugesan [14], they contain a large quantity of vitamin D_2_ precursors. The incorporation of mushroom leftover powder in a geriatric formulation is beneficial as it can increase the vitamin D content and help meet the vitamin D requirements of older adults.

Jaggery contains significant amounts of minerals (calcium, magnesium, potassium, etc.) and vitamins (vitamin A, vitamin B_1_, and vitamin B_2_). The micronutrients in jaggery provide a variety of health benefits, including antitoxin and anticarcinogenic properties. Jaggery has demonstrated that it is better than white sugar. Jaggery is well-recognized for generating heat and providing body energy [15].

Therefore, the objective of this study was to develop a geriatric formulation using leftover mushrooms, lactose-free milk powder, jaggary (sugar source), and oats. The developed formulation was rich in vitamin D_2_ and fibre. This formulation is lactose-free while providing the goodness of milk protein, is fibre-rich, and provides a significant amount of vitamin D to the geriatric population.

## 2. Materials and Methods

### 2.1. Materials

Lactose-free milk was from Amul, Gujrat, India; rolled oats was from Quaker Oats, PepsiCo Holdings India PVT. LTD., Gurgaon, India; mushroom leftovers (*Agaricus bisporus*) were collected from the vegetable market in Dehradun, India; and jaggery powder was collected from a local market in Dehradun. Chemicals used were N-Hexane (Loba Chemical PVT. LTD, Maharashtra, India), nitric acid (Merck Life Sciences PVT.LTD, Mumbai, India), sodium hydroxide (Molychem, Mumbai, India), MacConkey agar (HiMedia laboratories, Maharashtra, India), copper sulphate, boric acid, sulphuric acid, and bromocresol green (Avator Performance Materials, Maharashtra, India).

### 2.2. Processing of Mushrooms

Mushroom leftovers were cleaned with potable water and then blanched for 3 min at 90 °C and dried under sunlight for 12 h in order to increase active vitamin D_2_ [16]. The mushrooms were ground to a fine powder in an electric grinder (Sujata, Delhi, India) and kept in an airtight container stored at room temperature (30 ± 2 °C) until further use.

### 2.3. Preparation of Lactose-Free Milk Powder

Lactose-free milk was dehydrated on induction (Prestige, India) at 110 °C until it became a thick semi-solid form. After that, it was spread as a thin layer over a steel tray and air dried for 6 h for complete drying in a tray drier (Kerone, Mumbai, India). The dried milk was ground to a fine powder and then stored at room temperature (30 ± 2 °C) in a sealed container for further use.

### 2.4. The Preparation and Formulation of the Product

The product was developed by mixing lactose-free milk powder, mushroom powder, oat powder, and jaggery powder (shown in Figure 1) along with flavour (oil-based banana flavour) in different ratios mentioned in Table 1 and coded as MOS 1, MOS 2, and MOS 3. The prepared powdered products were then sealed in an airtight container in different ratios and analysed for physico-chemical properties.

### 2.5. Sensory Analysis

The hedonic scale (9-points) was used by 30 panel members (who were informed about the test) from the School of Health Sciences and Technology (SOHST), UPES, Dehradun to evaluate the sensory attributes of the products after dissolving them in warm water (5 g of powder in 25 mL of warm water), including flavour, taste, texture, consistency, colour, and overall acceptability. The intensity of each attribute was rated on a nine-point hedonic scale based on its intensity (9—extremely liked; 8—very much liked; 7—moderately liked; 6—slightly liked; 5—neither liked nor disliked; 4—slightly disliked; 3—moderately disliked; 2—very much disliked; and 1—extremely disliked) [17].

### 2.6. A Proximate Analysis of the Product and Individual Ingredients

A proximate analysis of the ash (using a muffle furnace at 550 °C for 4 h), moisture (using the oven drying method at 110 °C for 4 h), fat (using the Soxhlet method with hexane), protein (using the Kjeldahl method), and carbohydrate (using the balance method) contents of the products with different ratios and the individual ingredients was carried out [18]. The amount of carbohydrates was calculated by the balance method, and the energy value was calculated by multiplying the fat, carbohydrate, and protein proportions by their respective physiological energy values and calculating the total sum of those products [19].

### 2.7. Microbial Content

Samples of the product underwent microbiological testing in accordance with Biva et al. [20] with some modification. In order to count the aerobic bacteria along with the coliform, PCA (Plate Count Agar) and MA (MacConkey agar) were plated onto fivefold serial dilutions (10^−5^) for every sample in peptone water (0.1%). The plates of PCA were kept 32 ± 2 °C for 48 h. The MA plates were cultured for the counting of coliforms and kept at 32 ± 2 °C for 24 h. Colony-forming units (cfus), along with the most likely number, were used to count all bacteria and Gram-negative coliforms, respectively.

To count the bacterial cells that were isolated, the following formula was used:$$C.F.U=\frac{No.\;of\;colonies\;observed\times dilution\;factor}{Volume\;of\;sample\;taken}$$

### 2.8. Mineral Analysis

A total of 800 mg of incinerated sample was weighed and transferred into a volumetric flask followed by the addition of 3 mL of HNO_3_ and 3 mL of H_2_O_2_ (30% *v*/*v*). The samples were then heated to 180 °C for 10 min and allowed to cool before being diluted with 15 m of distilled water [21]. A mineral analysis of the product was carried out by using ICP-OES (Plasma Quant, PQ 9000, Jena, Germany).

### 2.9. FTIR Analysis

The samples were analysed using a Fourier-transform infrared (FT-IR) spectrometer (Perkin Elmer, Frontier FT-IR, Waltham, MA, USA) at infrared spectra (500 to 4000 cm^−1^). Pallets were made by mixing 400 mg of potassium bromide and 7 mg of the sample followed by applying 533.29 Pa of pressure. The samples were placed in an FT-IR chamber, and the IR probe was used to create the graph of functional groups.

### 2.10. GC-MS Analysis

The procedure to extract ergosterol from the samples for the GC-MS analysis was carried out as per Hossain and Goto [22]. In a 100 mL Erlenmeyer flask, 10 g of sample and 40 mL of methanol were shaken for 60 min at 33.51 rad/s using a horizontal shaker (SA-31, Yamato, Tokyo, Japan). After Whatman No.2 filtration, 10 mL of the filtrate was transferred to a 200 mL separating funnel, where 10 mL of a 3% KCl aqueous solution was added and thoroughly mixed. The liquid was shaken briskly by hand for 3 min after adding 10 mL of hexane. Finally, 1 L of the upper 1 mL of the hexane layer was injected using an on-column GC-MS injection device into a GC vial.

### 2.11. XRD Analysis

The solubility of the prepared powdered supplement was determined by using the X-ray diffraction technique (Bruker, D8 Advance Eco, Bruker corporation, Billerica, MA, USA). XRD is an important analysis method for the identification of food material based on different diffraction peak patterns. It is also useful for the phase identification of novel food products, where it yields information on how the actual structure varies from the standard one. Crystals are the regular forms of atoms; they are considered waves of electromagnetic radiation [23].

### 2.12. Experimental Setup and Statistical Analysis

The investigation was carried out in three repetitions using a completely randomized approach. A one-way analysis of variance was used to assess the data (ANOVA). The results were represented as the mean ± SD (standard deviation), with *p* < 0.05 denoting significance.

## 3. Results and Discussions

### 3.1. Proximate Analysis of Lactose-Free Milk Powder, Oat Powder, and Mushroom Powder

A proximate analysis was conducted for the lactose-free milk powder, oat powder, and mushroom powder. The amounts of moisture, fat, ash, protein, fibre, and carbohydrates in the lactose-free milk powder, oats, mushrooms, and jaggery are shown in Table 2. The amount of nutrient constituents that exist in the lactose-free milk powder falls within the range mentioned in earlier studies [24,25]. The percentage range of each constituent present in the oat powder was also reported in previous studies [26,27,28], and our results are concordant with their results. The nutrient compositions of the mushroom powder and jaggery were almost similar to the results of Farzana [29,30,31,32]. The moisture content was significantly high (*p* < 0.05) in the jaggery, and all four ingredients showed significant differences (*p* < 0.05) from each other. The lactose-free milk powder showed the lowest moisture content. The fat content was significantly different (*p* < 0.05) in all ingredients; it was the highest in the lactose-free milk powder and the lowest in the jaggery. The ash content of mushroom powder was significantly higher (*p* < 0.05), followed by lactose-free milk powder, jaggery, and oat powder. The protein content was significantly different (*p* < 0.05) in all ingredients; it was higher in mushroom powder and the lowest in jaggery. Fibre was absent in the lactose-free milk powder and jaggery, whereas the mushroom powder had a significantly higher fibre content than the oats. All ingredients had significantly different (*p* < 0.05) carbohydrate contents, with the highest being in the jaggery and the lowest being in the lactose-free milk powder.

### 3.2. Chemical Composition of Developed Products with Different Ingredient Ratios

#### 3.2.1. Moisture

The moisture content present in the three products was analysed. MOS 1 had a significantly higher (*p* < 0.05) moisture content than MOS 3, whereas MOS 2 showed a non-significant difference (*p* > 0.05) to both MOS 1 and MOS 3, respectively (Table 3). The lower moisture percentage of the products could be attributed to the drying process of each of the individual ingredients. Increased moisture encourages the growth of bacteria, which eventually degrades quality, making the moisture content a crucial component in maintaining food quality. The moisture content is a crucial component in the growth of microorganisms [33]. A moisture content below 8% suppresses microorganism growth, while some microbes can progressively multiply with an increase of more than 18% in moisture. Along with this, Wakeel [34] asserted that for dry or dehydrated materials, a 10% lesser moisture content is considered good for maintaining quality.

#### 3.2.2. Ash

The ash content present in the MOS 1, MOS 2, and MOS 3 products are shown in Table 3. The ash percent of MOS 3 was significantly (*p* < 0.05) higher than that of MOS 1, whereas MOS 2 showed a non-significant difference (*p* > 0.05) compared to both MOS 1 and MOS 3, respectively (Table 3). The higher ash level in MOS 3 may be due to the increased percentage of mushrooms in MOS 3, which enhanced the product’s mineral content as mushrooms are good suppliers of minerals [35]. Similar ash values were determined by Singh [36], where it was found that the ash content was higher because of the addition of mushroom powder to the product.

#### 3.2.3. Protein

The protein contents of MOS 1, MOS 2, and MOS 3 are presented in Table 3. The protein content of MOS 3 was significantly higher (*p* < 0.05) than those of MOS 2 and MOS 1, respectively. The reason for this may be due to the higher ratios of mushroom powder in the former samples, as mushrooms contain a high quantity of protein, as reported by Salehi [36].

#### 3.2.4. Fat

The percentages of fat of the samples for MOS 1, MOS 2, and MOS 3 are shown in Table 3. A non-significant difference (*p* > 0.05) was observed between all samples. MOS 1 showed the highest fat content, which might be due to the highest ratio of oat powder in MOS 1, which contributes to the slightly increased fat content in the sample, as the fat content of oats is higher than that of mushrooms, as shown in Table 2. Aly [36] and Farzana [29] described that the fat percentages of oats and mushrooms are 5.44 and 2.5%, respectively, which shows that oats have a higher amount of fat than mushrooms.

#### 3.2.5. Fibre

The fibre contents of the samples for MOS 1, MOS 2, and MOS 3 are shown in Table 3. MOS 3 showed a significantly highest (*p* < 0.05) fibre content than MOS 2 and MOS 1, respectively. This is due to an increase in the concentration of mushroom powder in the samples, as mushrooms have the highest percentage of fibre. Zhao et al. [37] also reported that mushrooms are a good source of fibre and they can increase the fibre content for food fortification. Farzana et al. [29] also reported that mushrooms contain a good amount of fibre and reported that they have 12.5% of fibre. According to Rebello et al. [38], oats are also a good source of fibre and contribute to the fibre content in food. Youssef et al. [38] found in their study that oats contain 3.53% of fibre, which supports our findings. Milk and milk products are fibre-free [39].

#### 3.2.6. Carbohydrate

The carbohydrate contents were found for MOS 1, MOS 2, and MOS 3 and are shown in Table 3. MOS 1 has the largest carbohydrate content when compared to the other two products. This could be due to the high oat-to-mushroom powder ratio in MOS 1, as oats have a higher carbohydrate content than mushrooms (Table 2). Krishnamoorthy et al. [31] reported that there is 41.6% of carbohydrates in mushrooms, and Aly [26] reported that there is 66.89% of carbohydrates in oats. Similarly, in Table 2, it is shown that the carbohydrate content of oats is higher than that of mushrooms, indicating that our findings are consistent with the literature.

#### 3.2.7. Energy Value

The energy values obtained from the study for MOS 1, MOS 2, and MOS 3 are shown in Table 3. MOS 1 showed a significantly (*p* < 0.05) higher energy value than MOS 3, whereas MOS 2 showed a non-significant (*p* > 0.05) difference compared to MOS 1 and Mos 3, respectively. The lowest energy value was found for MOS 3, which may be due to the lower fat, protein, and carbohydrate contents present because of the higher ratio of mushroom powder than oat powder. Similar energy values were observed in the studies of Bembem and Agrahar-Murugkar [40] and Bharti et al. [40].

### 3.3. Organoleptic Evaluations of the Developed Samples

The results show that MOS 3 was liked the most for all sensory parameters in comparison to MOS 1 and MOS 2, respectively. In flavour and taste, MOS 3 showed higher (*p* < 0.05) sensory scores. The texture, colour, and consistency of all three samples were statistically (*p* > 0.05) similar. The overall acceptability scores of MOS 1, MOS 2, and MOS 3 were 6.8, 7.2, and 7.5, respectively, as shown in Figure 2. MOS 3 showed the highest sensory acceptability compared to the other two products (Table 4). Bembem and Agrahar-Murugkar [40] and Singh et al. [41] also reported similar sensory scores (ranges from 6–8) in their geriatric formulations.

### 3.4. Microbial Analysis of Developed Products with Different Ratios

The bacterial cfu counts for MOS 1, MOS 2, and MOS 3 were 3 × 10^2^, 3 × 10^3^, and 3 × 10^2^ cfu/g, respectively (Table 4). The coliform counts for MOS 1, MOS 2, and MOS 3 were absent. The results showed that the product’s viable bacteria are within the safe limit [42]. The developed product’s microbial range was within safe limits (4 × 10^4^ cfu/g for dried fruits and vegetables and 30 × 10^3^ cfu/g for milk powder) according to the FSSAI, 2011. The results show that the viable bacteria in the product are within the limits of safety [42]. Additionally, no Coliform sp. was detected in any of the samples, indicating that they are all microbiologically safe, as reported elsewhere [43,44].

### 3.5. Mineral Analysis

Minerals are essential parts of our diet. They perform a wide range of tasks, including serving as the building blocks for our bones, affecting muscle and nerve activity, etc. [45]. For most people, a balanced diet is adequate to give the body the necessary levels of essential minerals. However, a growing number of people are now at risk of mineral deficiencies. Most of these are older people [46]. The amounts of Ca, K, Mg, Na, Zn, Cu, and Fe present in MOS 1, MOS 2, and MOS 3 are shown in Table 5. Pb, Ag, As, Hg, Co, Mn, and Ni were not present.

All samples showed non-significant (*p* > 0.05) differences in the potassium and sodium contents. In comparison to MOS 1 and MOS 2, MOS 3 had the highest concentrations of Ca, Mg, Zn, Cu, and Fe. MOS 3 had the lowest levels of K and Na compared to the other two samples.

### 3.6. FTIR Analysis of Developed Products with Different Ratios

The FTIR spectra of the three samples (MOS 1, MOS 2, and MOS 3) are presented in Figure 3. No change in the infrared absorption peak shape of the three samples was detected. The peaks at 3352.00 cm^−1^, 3350.00 cm^−1^, and 3243.00 cm^−1^ of MOS 1, MOS 2, and MOS 3 show O-H stretching (Figure 3). The peaks at 1745.83 cm^−1^, 1745.87 cm^−1^, and 1745.67 of MOS 1, MOS 2, and MOS 3 show C=O stretching [47]. The peaks at 1648.78 cm^−1^, 1648.00, and 1647.00 cm^−1^ of MOS 1, MOS 2, and MOS 3 indicate C=C stretching. The peaks at 2928.00 cm^−1^ and 2932.00 cm^−1^ for MOS 1 and MOS 2 relate to C-H stretching. The peak at 2854.06 cm^−1^ was higher for MOS 1 compared to that for MOS 2 and MOS 3, indicating the presence of an aldehyde group. The peak at 2854.06 cm^−1^ in the FTIR analysis was higher for MOS 1 than for MOS 2 and MOS 3, indicating the presence of an aldehyde group (Figure 3). This could be due to the presence of a high ratio of oat powder (the presence of hexanal, 2-butanal, etc.) in MOS 1 compared to the other two. In addition, the samples contain glucose and galactose as a result of lactose breakdown in lactose-free milk, which may result in the presence of aldehyde groups. Along with this, jaggery, mushroom, and oats also contribute to the increase in free sugar in the sample.

### 3.7. XRD Analysis of Developed Products with Different Ratios

The different diffraction peaks show different crystalline structures in the samples in Figure 4. The XRD analysis showed that there was an increase in the crystalline structure in MOS 1 (Figure 4), which may be the result of the milk powder as it contains lactose. In MOS 1 and MOS 2, the ratios of oat powder were, respectively, higher in comparison to that of MOS 3, as oats contain a higher amount of starch and mushroom contains soluble sugar; this might result in a decrease in crystallinity and an increase in solubility in MOS 3. Therefore, higher crystallinity was observed in MOS 1 in comparison to MOS 2 and MOS 3. There are limited data available on oats and mushroom formation regarding their solubility and crystallinity.

### 3.8. GC-MS Analysis

The recognition of ergosterol was based on the interpretation of the mass spectra and by comparing the data of the retention time and mass spectra with those of Hull et al. [48]. The retention time is similar and shows the presence of ergosterol. It was found that with the increase in mushrooms in the sample, the peak area of ergosterol also increased (MOS 1—2,895,734,784.0; MOS 2—3,143,375,104.0; and MOS 3—6,110,513,664.0), as shown in Figure 5. Extensive investigations have been carried out, and it was found that ergosterol can be transformed into vitamin D_2_ by exposure to either artificial or natural UV rays [49,50]. In the geriatric population, vitamin D deficiency is the most common problem. According to reports, low serum vitamin D levels are prevalent health issues for older adults and are linked to decreased physical performance, an increase in the likelihood of falls, and a higher risk of fractures. Additionally, they are also likely to experience cognitive decline, despair, and anxiety [51].

## 4. Conclusions

Based on biochemical and sensory analyses, MOS 3 is the most acceptable of the three products. The product’s bacterial count is within safe limits, which is attributed to its low moisture content. It is also acceptable from a microbiological viewpoint. Furthermore, the product has high protein, fibre, and ash contents, indicating the presence of high mineral contents, including Ca, K, Mg, Na, Zn, Cu, and Fe (6481, 8059, 76.28, 1864, 5.46, 0.31, and 0.35 mg/kg, respectively) with fewer calories. In addition, the product contains a high amount of ergosterol. All of the products are microbiologically safe with a limited number of microbial counts. All of these factors make the formulated product a good choice to fulfil nutritional demand, especially in the geriatric population. This could significantly improve the older population’s nutritional status. Further studies can be conducted on this developed product like in vivo studies, in vitro studies, toxicological tests, etc. Also, its bioavailability can be checked using animal models.

## Figures and Tables

**Figure 1 foods-13-01738-f001:**
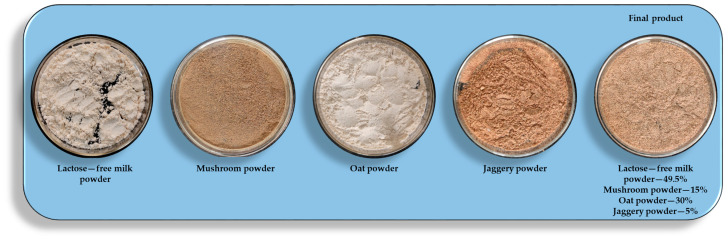
Illustration of different ingredients and final product.

**Figure 2 foods-13-01738-f002:**
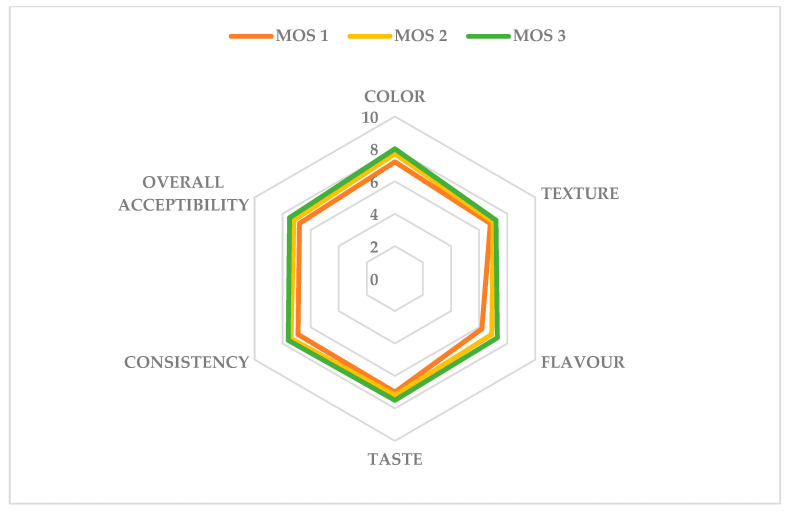
Graph showing sensory scores of samples.

**Figure 3 foods-13-01738-f003:**
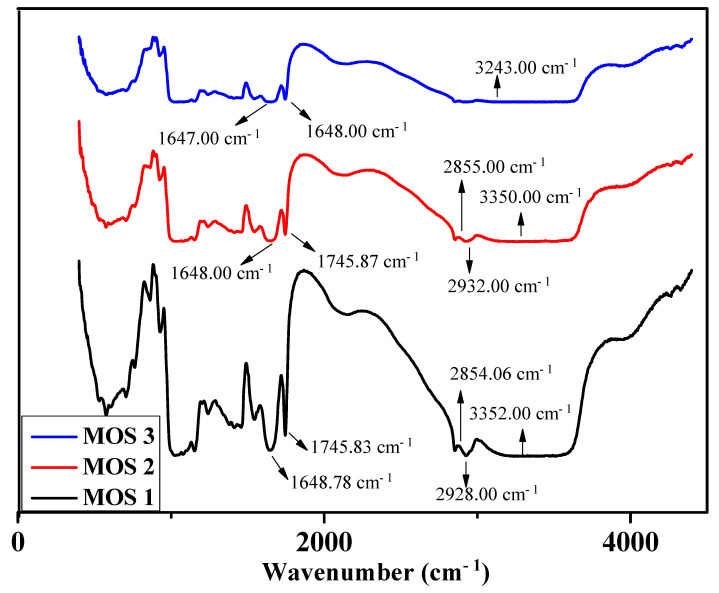
FTIR spectra of the developed product.

**Figure 4 foods-13-01738-f004:**
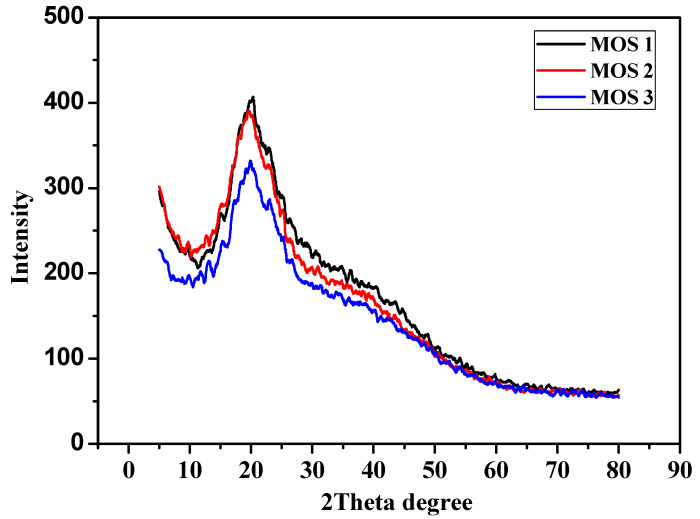
XRD of developed products (MOS 1, MOS 2, and MOS 3).

**Figure 5 foods-13-01738-f005:**
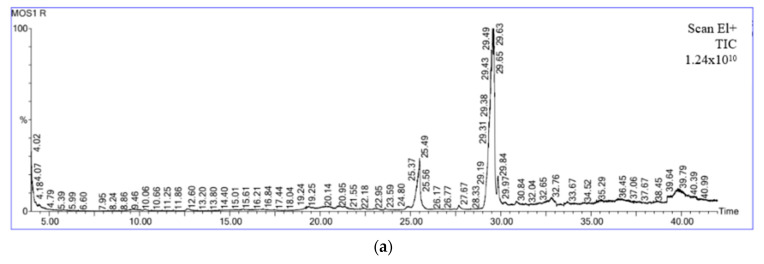
GC-MS analysis of product for ergosterol: (**a**) MOS 1, (**b**) MOS 2, and (**c**) MOS 3.

**Table 1 foods-13-01738-t001:** The amount of ingredients used for the preparation of the formulation.

Ingredients	Amount (%)		
	MOS 1	MOS 2	MOS 3
Lactose-free milk powder	49.5%	49.5%	49.5%
Mushroom powder	5%	10%	15%
Oat powder	40%	35%	30%
Jaggery powder	5%	5%	5%
Flavours	0.5%	0.5%	0.5%

**Table 2 foods-13-01738-t002:** Proximate analysis of lactose-free milk powder, oats, and mushroom powder.

Constituents (%)	Lactose-Free Milk Powder	Oat Powder	Mushroom Powder	Jaggery
Moisture	2.36 ± 0.01 ^a^	9.16 ± 0.02 ^c^	4.30 ± 0.40 ^b^	23.04 ± 0.02 ^d^
Fat	27.03 ± 0.01 ^d^	5.86 ± 0.005 ^c^	2.43 ± 0.05 ^b^	0.12 ± 0.04 ^a^
Ash	5.63 ± 0.04 ^b^	1.48 ± 0.01 ^a^	7.18 ± 0.17 ^c^	2.02 ± 0.05 ^a^
Protein	26.01 ± 0.02 ^c^	12.62 ± 0.05 ^b^	31.00 ± 0.26 ^d^	1.05 ± 0.01 ^a^
Fibre	0.00 ± 0.00 ^a^	3.57 ± 0.08 ^b^	12.56 ± 0.05 ^c^	0.00 ± 0.00 ^a^
Carbohydrate	38.94 ± 0.01 ^a^	67.28 ± 0.34 ^c^	42.52 ± 0.40 ^b^	73.77 ± 0.23 ^d^

The values are written in the form of the mean of the data ± SD. The different subscript letters ^a–d^ represent a significance (*p* < 0.05) difference in the column.

**Table 3 foods-13-01738-t003:** Proximate analyses of developed products with different ingredient ratios.

Constituents (%)	MOS 1	MOS 2	MOS 3
Moisture	6.13 ± 0.15 ^b^	5.89 ± 0.25 ^ab^	5.64 ± 0.20 ^a^
Fat	15.84 ± 0.25 ^a^	15.67 ± 0.10 ^a^	15.49 ± 0.15 ^a^
Ash	3.84 ± 0.07 ^a^	4.12 ± 0.15 ^ab^	4.34 ± 0.36 ^b^
Protein	19.54 ± 0.17 ^a^	20.46 ± 0.20 ^b^	21.38 ± 0.24 ^c^
Fibre	2.04 ± 0.08 ^a^	2.56 ± 0.20 ^b^	2.95 ± 0.10 ^c^
Carbohydrate	52.07 ± 0.27 ^c^	50.73 ± 0.30 ^b^	49.49 ± 0.80 ^a^
Energy (kcal/100 g)	429.03 ± 1.00 ^b^	425.79 ± 1.74 ^ab^	422.89 ± 1.50 ^a^

The values are written in the form of the mean of the data ± SD. The different subscript letters ^a–c^ represent a significance (*p* < 0.05) difference in the column.

**Table 4 foods-13-01738-t004:** Microbiological and sensory evaluations of developed products with different ingredient ratios.

Sample	Viable Count		Sensory Evaluation
	Bacteria (cfu/g)	Coliforms (mpn)	Overall Acceptability (Out of 9)
MOS 1	3 × 10^2^	Nil	6.8 ± 0.02
MOS 2	3 × 10^3^	Nil	7.2 ± 0.01
MOS 3	3 × 10^2^	Nil	7.5 ± 0.02

The values are written in the form of the mean of the data ± SD.

**Table 5 foods-13-01738-t005:** The amounts of minerals present (mg/kg) in the product.

Sample	Mineral (mg/kg)													
	Ag	As	Ca	Hg	K	Mg	Na	Pb	Zn	Co	Cu	Fe	Mn	Ni
MOS 1	ND	ND	6303.2 ± 0.03 ^a^	ND	8158.0 ± 0.09 ^a^	75.37 ± 0.03 ^b^	1866 ± 0.20 ^a^	ND	4.95 ± 0.005 ^a^	ND	0.19 ± 0.004 ^a^	0.26 ± 0.002 ^b^	ND	ND
MOS 2	ND	ND	6392.0 ± 0.09 ^b^	ND	8108.0 ± 0.20 ^a^	73.79 ± 0.09 ^a^	1865 ± 0.03 ^a^	ND	5.08 ± 0.002 ^a^	ND	0.16 ± 0.003 ^a^	0.12 ± 0.001 ^a^	ND	ND
MOS 3	ND	ND	6481.0 ± 0.20 ^c^	ND	8059.0 ± 0.03 ^a^	76.28 ± 0.20 ^c^	1864 ± 0.03 ^a^	ND	5.46 ± 0.10 ^b^	ND	0.31 ± 0.003 ^b^	0.35 ± 0.007 ^c^	ND	ND

The values are written in the form of the mean of the data ± SD. ^a–c^ Means within columns with different superscript are significantly different (*p* < 0.05) from each other.

## Data Availability

The original contributions presented in the study are included in the article/supplementary material, further inquiries can be directed to the corresponding author/s.

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
