# Peer review of "The Utilisation of Mushroom Leftovers, Oats, and Lactose-Free Milk Powder for the Development of Geriatric Formulation"

_foods, 2024, doi:10.3390/foods13111738_

Round 1

Reviewer 1 Report

Comments and Suggestions for Authors

The approach is simple to begin with, but it could work. However, as stated in the introduction, one of the problems during ageing is the loss of efficiency of enzyme systems.

This leads to lactose intolerance, but also to other sugars such as trehalose found in mushrooms.

Lactose intolerance is well studied and milk is a more widely consumed food than mushrooms, but there are data indicating that at least 1:20,000 people are allergic to trehalose, taking into account the minority consumption of mushrooms compared to milk is very important rate.

In this respect, I have my doubts that the initial approach with the elderly as the target population I am not really sure if it is the most appropriate. I think you must add this information and some bibliography with this topics at least in the introduction.

It is true, on the other hand, that the work is interesting as a starting point and, as the conclusions indicate, it requires further studies to prove it.

But I wanto to provide a chance of publication. Although it is true that, from my point of view, it needs an intensive revision, mainly of the discussion, which is very simple and evident, but at the same time it is difficult to read due to the lack of connection between paragraphs.

Moreover, it lacks uniformity in the units. The use of the international system of units is highly recommended and some of the methods used are obsolete or can be justified with more updated bibliography.

Author Response

Thanks for your review and time. I have made corrections as per your suggestion. The correction are made and answered in the attached document.

Reviewer comment-1

The approach is simple to begin with, but it could work. However, as stated in the introduction, one of the problems during ageing is the loss of efficiency of enzyme systems.

This leads to lactose intolerance, but also to other sugars such as trehalose found in mushrooms.

I agree with the viewpoint regarding the trehalose found in the mushroom which needed to get attention. But also, the amount of mushroom (5-15%) used in the formulation is less than that of other constituents present in the formulation which might decrease the chance of problems face by trehalose intolerant. Also we have developed this formulation particularly for the lactose intolerant geriatric population.

Lactose intolerance is well studied and milk is a more widely consumed food than mushrooms, but there are data indicating that at least 1:20,000 people are allergic to trehalose, taking into account the minority consumption of mushrooms compared to milk is very important rate.

In this respect, I have my doubts that the initial approach with the elderly as the target population I am not really sure if it is the most appropriate. I think you must add this information and some bibliography with this topics at least in the introduction.

Line 77-82- The information regarding trehalose and trehalose intolerance has been added in the introduction and highlighted with yellow.

It is true, on the other hand, that the work is interesting as a starting point and, as the conclusions indicate, it requires further studies to prove it.

But I want to provide a chance of publication. Although it is true that, from my point of view, it needs an intensive revision, mainly of the discussion, which is very simple and evident, but at the same time it is difficult to read due to the lack of connection between paragraphs.

Line 195-200, 218-226, 231-234, 243-248, 253-259, 264-267, 273-275, 285-288, 295-299, 332-338, 345-350- All the paragraphs in discussion part have been merged with the results as per the journal guidelines and highlighted with yellow.

Moreover, it lacks uniformity in the units. The use of the international system of units is highly recommended and some of the methods used are obsolete or can be justified with more updated bibliography.

Line 147, 150, 167, 172- SI units have been uniformed throughout the paper, bibliography has been updated for some methods and highlighted with yellow.

What kind of training have the panelist received? Describe or cite the ISO standard or reference you have used for this purpose?

Line 131, 138- The word trained has been removed and instead of that “instructed about the test” has been added and highlighted with yellow. The panelists were instructed about the test before the sensory evaluation. Updated reference has been added supporting the usage of 9-points hedonic scale for sensory evaluation for product.

Line 134-references rather out of date, check if in any later edition there are no changes in methodology and cite more up to date

Line 143- The reference has been updated and highlighted with yellow.

Line 140- Very old method

Line 149- updated reference has been added and highlighted with yellow

Line 151- mL: be homogeneous throughout the text

mL has been homogenized throughout the text

Line 151- Correct formulation with subscripts

Formulation with subscript has been corrected throughout the manuscript and highlighted with yellow

Line 152- °C separate: °180 C

Line 160- Correction has been made and highlighted with yellow

Line 158-units separate

Line 166- Correction has been made and highlighted with yellow.

Line 159- Units in International System

Line 167- Unit mmHg has been changed to Pa (SI unit) and highlighted with yellow

Line 164-rpm is not international system unit

Line 172- 320 rpm has been changed to 33.51 rad/s (SI unit) and highlighted with yellow

Why does it represent up to 30 in the graph if there are no values above 9. This makes the differences less visible.

The sensory evaluation graph has been corrected

They did an FTIR which helps to see which of the products has more crystals or is more soluble but they didn't do a solubility study, I think it would have been wise to do that.

The suggestion of a solubility study is appreciable and noted for the future.

The discussion lacks connectors between paragraphs, Not all the ideas are interconnected which makes the discussion hard to read

Line 195-200, 218-226, 231-234, 243-248, 253-259, 264-267, 273-275, 285-288, 295-299, 332-338, 345-350- All the paragraphs in discussion part have been merged with the results as per the journal guidelines and highlighted with yellow.

Reviewer 2 Report

Comments and Suggestions for Authors

This paper studies a new formulation for the geriatric population. The topic is not a novel research but its results can be applicable. Some comments and questions should be answered as follows.

Detailed comments;

·         The authors study the effect of different amounts of mushroom and oats powders on the final product. So, I propose the lactose-free milk powder must be deleted from the title of the manuscript.

·         Keywords should be arranged alphabetically.

·         Please show an image of Jaggery as well as the final product.

·         What about the safety of this plant?

·         Which flavor did you use? Is it powder or oil?

·         In sensory evaluation, how did you analyze the texture and consistency of the final powder?

·         According to journal guidelines, results and discussion should be merged.

·         Duncan's analysis is wrong. The large numbers should be marked using the first letters.

·         Sensory evaluation should be corrected. Present the attributes on polygon vertices.

Regards,

Author Response

Thanks for your review and time. I have made all the corrections as suggested. The corrections made are presented in attached document.

Reviewer comments-2

  • The authors study the effect of different amounts of mushroom and oats powders on the final product. So, I propose the lactose-free milk powder must be deleted from the title of the manuscript.

The proposal for the title change is appreciable, but the title includes major constituents of the developed product including lactose-free milk powder. The title is supposed to reflect the major constituents present and not the effect of different amounts, although the study was conducted on different proportions but the aim was to develop a product with all the mentioned constituents in the title. 

  • Keywords should be arranged alphabetically.

Line 34- Keywords have been arranged alphabetically and highlighted with yellow.

  • Please show an image of Jaggery as well as the final product.

Line 129- Image of final product and every ingredients has been added and highlighted with yellow.

  • What about the safety of this plant?

All the plant based ingredients used in the formulation have GRAS status, therefore no safety issues are there.

  • Which flavor did you use? Is it powder or oil?

Line 123- Oil based banana flavor was used. It has been mentioned in the manuscript and highlighted with yellow.

  • In sensory evaluation, how did you analyze the texture and consistency of the final powder?

Line 133- The sensory evaluation of the powder product was analysed after dissolving it in warm water (5g powder in 25 ml of warm water). The line has been mentioned in the manuscripts and highlighted with yellow.

  • According to journal guidelines, results and discussion should be merged.

Line 195-200, 218-226, 231-234, 243-248, 253-259, 264-267, 273-275, 285-288, 295-299, 332-338, 345-350- All the paragraphs in discussion part have been merged with the results as per the journal guidelines and highlighted with yellow.

  • Duncan's analysis wrong. The large numbers should be marked using the first letters.

Corrections has been made and highlighted with yellow

  • Sensory evaluation should be corrected. Present the attributes on polygon vertices.

Graph showing sensory evaluation has been corrected and the attributes has been presented on polygon vertices.

Round 2

Reviewer 2 Report

Comments and Suggestions for Authors

the changes are acceptable.